# Targeting the Gut–Brain Axis with Plant-Derived Essential Oils: Phytocannabinoids and Beyond

**DOI:** 10.3390/nu17091578

**Published:** 2025-05-03

**Authors:** Luca Camarda, Laura Beatrice Mattioli, Ivan Corazza, Carla Marzetti, Roberta Budriesi

**Affiliations:** 1Department of Pharmacy and Biotechnology (FaBiT), Food Chemistry and Nutraceutical Lab, Alma Mater Studiorum-University of Bologna, Via Belmeloro 6, 40126 Bologna, Italy; laurabeatrice.mattioli@unibo.it (L.B.M.); roberta.budriesi@unibo.it (R.B.); 2Health Sciences and Technologies-Interdepartmental Center for Industrial Research (CIRI-SDV), Alma Mater Studiorum-University of Bologna, 40126 Bologna, Italy; ivan.corazza@unibo.it; 3Department of Medical and Surgical Sciences (DIMEC), Alma Mater Studiorum-University of Bologna, 40138 Bologna, Italy; 4Valsambro S.r.l., 40121 Bologna, Italy; carla.marzetti@valsambro.it

**Keywords:** gut–brain axis, essential oils, endocannabinoid system, cannabis, microbiota, neuroprotection

## Abstract

**Background**: The gut–brain axis (GBA) is a complex bidirectional communication system that links the gastrointestinal tract and the central nervous system. Essential oils (EOs) have emerged as promising natural compounds capable of modulating this axis. **Methods**: A comprehensive analysis of the recent literature was conducted, focusing on studies investigating the effects of EOs on the GBA. Particular attention was given to the endocannabinoid system, the role of cannabis-derived EOs, and other plant-based EOs with potential neuroprotective and gut microbiota-modulating effects. **Results**: Among the EOs analyzed, cannabis essential oil (CEO) gained attention for its interaction with cannabinoid receptors (CBR1 and CBR2), modulating gut motility, immune responses, and neurotransmission. While acute administration of the CEO reduces inflammation and gut permeability, chronic use has been associated with alterations in gut microbiota composition, potentially impairing cognitive function. Other EOs, such as those from rosemary, lavender, eucalyptus, and oregano, demonstrated effects on neurotransmitter modulation, gut microbiota balance, and neuroinflammation, supporting their potential therapeutic applications in GBA-related disorders. **Conclusions**: EOs demonstrate promising potential in modulating the GBA through mechanisms including neurotransmitter regulation, gut microbiota modulation, and anti-inflammatory activity. At the same time, phytocannabinoids offer therapeutic value; their long-term use warrants caution due to potential impacts on microbiota. Future research should aim to identify EO-based interventions that can synergistically restore GBA homeostasis and mitigate neurodegenerative and gastrointestinal disorders.

## 1. Introduction

In recent years, the growing incidence of mental health conditions and neurodegenerative diseases has been consistently reported in the scientific literature, driven by a range of contributing factors. Epidemiological data show a marked increase in disorders like anxiety, depression, and bipolar disorder, mainly affecting younger age groups [1]. At the same time, diseases like Alzheimer’s and Parkinson’s are especially prevalent among the elderly [2,3].

The global increase in gastrointestinal diseases has been primarily linked to lifestyle transitions, dietary shifts, and exposure to environmental stressors. Recent studies indicate a significant rise in conditions such as Inflammatory Bowel Disease (IBD), Irritable Bowel Syndrome (IBS), and metabolic-associated fatty liver disease [4]. Furthermore, the incidence of Crohn’s disease (CD) and ulcerative colitis (UC) has surged globally, particularly in urbanized regions, reflecting a shift in gut microbiota composition due to Westernized diets and increased antibiotic use [5]. Heavy metal exposure and environmental toxins have also been linked to the growing prevalence of IBD, highlighting the role of external pollutants in gut dysbiosis [6].

Due to the increasing prevalence of diseases affecting the central nervous system (CNS) and the gut, it is reasonable to assume a close connection between these two systems. This hypothesis is well-supported in the current literature and is central to research into the gut–brain axis (GBA) [7]. Consequently, researchers are increasingly challenging the traditional separation between gut and brain pathophysiology, proposing a more integrated approach [8,9].

The GBA represents a complex communication network between the gastrointestinal (GI) tract and the CNS. The GBA involves bidirectional communication between the gut microbiota, the Enteric Nervous System (ENS), and the CNS [10]. Research conducted in recent years has shown that dysregulation of this axis is implicated in various conditions, including IBS, depression, and anxiety [11]. Recently, functional GI disorders have become better known as Disorders of Gut–Brain Interaction (DGBIs); these disorders are a group of non-structural abnormalities, including IBS and chronic gastrointestinal diseases. The pathophysiology of DGBIs are associated with psychological stress, diet, and alterations in microbiota. DGBIs indicate the brain–gut connection, complex interplay between the central and peripheral nervous systems, and GI alterations [12].

Understanding the importance of the GBA has significant implications for managing DGBIs, neuropsychiatric disorders, and age-related cognitive decline [13].

In recent years, natural extracts and nutraceuticals have garnered attention for their health benefits, particularly in modulating gut microbiota and influencing brain function [14]. For example, non-absorbable dietary components metabolized by the gut microbiota have been associated with improved emotional and cognitive function, reinforcing the concept of Nutritional Psychiatry [15].

Furthermore, emerging research strongly suggests that dietary interventions, including various food or plant extracts and multiple essential oils (EOs), can modulate the GBA, impacting mental health, neurological function, and GI health [15]. These interventions affect gut microbiota composition and influence the production of metabolites, neurotransmitters, and hormones that directly impact the brain [16,17].

Numerous compounds sourced from foods, plants, and food waste by-products have demonstrated promising effects on the GBA [18,19].

EOs, known for their volatility and plant origin, have attracted growing interest for their diverse biological effects, including antioxidant, antimicrobial, and anti-inflammatory effects [20,21].

Recent studies have revealed the innovative potential of EOs as bioactive agents capable of modulating the GBA through multiple and precise mechanisms. One remarkable feature is their ability to cross biological membranes, including the blood–brain barrier (BBB), interacting with peripheral and central nervous system targets [22]. For example, oregano essential oil has been shown to interact with cannabinoid receptors, particularly CBR2, which are expressed not only in the CNS but also in immune and gastrointestinal tissues, suggesting a multi-level modulation of neuroimmune signaling [23].

EOs demonstrate pleiotropic biological activity due to their chemically diverse components, primarily terpenes and phenylpropanoids [24]. These compounds exert antioxidant, anti-inflammatory, antimicrobial, and cardiovascular protective effects, making them promising candidates for therapeutic applications beyond traditional uses [22,24,25]. In particular, compounds like thymol, carvacrol, β-caryophyllene, and linalool exhibit modulatory effects on neurotransmission, inflammation, and oxidative stress—key elements in GBA-related pathophysiology [22,24].

Recent developments have also emphasized the therapeutic promise of combining plant-based compounds with dietary fibers and polyphenols, especially those derived from agri-food waste. These bioactives can synergistically shape microbial composition while promoting anti-inflammatory and neuroprotective responses. As highlighted in the emerging literature, such integrated strategies enhance health interventions’ sustainability and open new frontiers for personalized and preventive approaches to gut–brain axis modulation, particularly in chronic and lifestyle-related disorders.

Moreover, EOs are increasingly being explored as dietary modulators of gut microbiota. Their antimicrobial and prebiotic-like properties shape gut microbial composition and thus indirectly influence the brain via microbial metabolites and immune signals [22]. Additionally, their vasorelaxant and antihypertensive effects further suggest a role in improving cerebral blood flow and endothelial function, which are relevant to cognitive function and neurovascular health [25].

These findings collectively highlight the innovative role of EOs as modulators of the GBA, acting on microbial ecosystems and peripheral tissues, while also exerting central neuromodulatory actions. The ability of certain EO constituents to act on CB2 and opioid receptors, as demonstrated in recent studies, represents a promising frontier in integrative and nutritional neuroscience [23].

The research on EOs has significantly expanded over recent decades, incorporating advanced extraction [26] and analytical techniques [27] and exploring diverse applications in medicine, agriculture, and the food industry [28,29].

## 2. Methods

This narrative review was conducted in three main phases: literature search, screening of abstracts and full texts, and discussion of results. To identify the relevant literature, the following databases were explored: PubMed, Scopus, Science Direct, Web of Science, and Google Scholar. The final search was completed in February 2025 and included peer-reviewed international articles, online reports, and electronic books published in English. The primary keyword used was “gut-brain axis”, combined with terms such as “essential oil”, “cannabinoids”, “neuroprotection”, “neurodegenerative”, “endocannabinoid system”, and “microbiota”. After the search, abstracts were reviewed to confirm their relevance to the topic. Duplicate entries were removed, and the remaining abstracts were assessed based on the inclusion criteria. Eligible studies were primarily those published within the past 10 years to ensure up-to-date scientific evidence. From the general literature screening, 7950 articles were initially identified; however, only about 1% met the inclusion and exclusion criteria and were deemed relevant for this review. As this is a narrative review, it was not necessary to formally register the search strategy on a specific platform.

## 3. Gut–Brain Axis Overview

### 3.1. Anatomy and Physiology of the Gut–Brain Axis

The GBA represents a complex signaling network that connects the gastrointestinal tract and the CNS. This bidirectional network is crucial for maintaining homeostasis and regulating both physical and mental health. The GBA encompasses multiple components, including the CNS, ENS, gut microbiota, and various pathways, such as the vagus nerve and neuroimmune signaling [30].

At the core of this axis is the CNS, which processes signals from the gut via neural pathways such as the vagus nerve and spinal afferents. The CNS influences gut function through descending signals that regulate motility, secretion, and immune responses [31]. Key neurotransmitters, including serotonin, GABA, and dopamine, play a role in these processes. Interestingly, much of the body’s serotonin is produced in the gut, with microbial involvement, highlighting the microbiota’s influence on brain chemistry [32].

Known as the “second brain”, the ENS functions semi-independently yet remains tightly connected to the CNS. Situated in the walls of the GI tract, the ENS comprises a vast network of neurons responsible for coordinating digestive processes, such as gut motility and secretion [33]. The brain processes sensory signals ascending from the gut and sends descending signals back to the gut through autonomic neurons. These neurons influence intestinal immune responses through local axon reflexes or neuronal circuits within the GBA. This neuroimmune interaction is crucial for maintaining gut homeostasis and resolving diseases [34].

The gut microbiota, composed of trillions of microorganisms, plays a central regulatory role in the functioning of the GBA. These microbes produce bioactive metabolites, such as short-chain fatty acids (SCFAs), which influence inflammation, immune function, and neuroimmune signaling. Dysbiosis, or an imbalance in gut microbial composition, has been implicated in a range of neurological and psychiatric conditions, including anxiety and depression [35]. Additionally, the gut microbiota communicates with the brain through immune pathways and the production of neurotransmitter precursors, further underscoring its role in the GBA [36].

The vagus nerve is a primary conduit for gut–brain communication. It transmits sensory information from the gut to the brain and motor signals back to the gut, influencing digestion, inflammation, and gut barrier integrity. This nerve is susceptible to changes in gut microbiota composition and activity, serving as a critical link between microbial signals and neural responses [37].

Moreover, the GBA integrates the neuroimmune and neuroendocrine systems, including the hypothalamic–pituitary–adrenal (HPA) axis, which regulates stress responses. Immune cells in the gut produce cytokines that can signal the brain, while the HPA axis influences gut function through stress hormones like cortisol. This dynamic interaction is vital for gut barrier function and immune regulation, and its disruption has been associated with conditions like IBD and metabolic disorders [38].

One of the most frequent symptoms observed in patients with gut–brain interaction disorders is visceral pain. Some essential oils, such as peppermint essential oil, are able to reverse visceral hypersensitivity and therefore represent a safe and effective therapeutic option [39]. 

### 3.2. Mechanisms of Interaction

The GBA relies on a complex network of bidirectional communication mechanisms that connect the GI tract and the CNS. These interactions occur through neural, hormonal, immune, and microbial pathways, ensuring a constant exchange of information to maintain homeostasis and respond to external stimuli [40].


**Neural Pathways**


One of the primary communication routes within the GBA is the neural pathway involving the vagus nerve and the ENS. The vagus nerve transmits sensory signals from the gut to the brain and motor signals from the brain to the gut [41]. It is pivotal in regulating gut motility, secretion, immune responses, and modulating systemic inflammation. In addition to the vagus nerve, other extrinsic nerve pathways, such as the pelvic nerves and the splanchnic nerves, also contribute to gut–brain communication. These pathways are particularly important in the transmission of visceral pain signals and play a critical role in the pathophysiology of gut–brain interaction disorders. The ENS, often termed the “second brain”, operates autonomously but collaborates with the CNS to process gut-derived sensory information and modulate gut activity [9].

2.
**Hormonal Signaling**


Gut hormones, produced by enteroendocrine cells scattered throughout the GI tract, act as critical mediators of gut–brain communication. These hormones include glucagon-like peptide-1 (GLP-1) [42] and ghrelin, influencing appetite regulation and energy metabolism [43]. Hormones like serotonin, synthesized mainly in the gut, affect CNS functions, including mood and cognition. Gut microbiota can also modulate the release of these hormones, demonstrating a dynamic interaction between microbes and the endocrine system [44].

3.
**Immune-Mediated Communication**


The immune system is a vital component of the GBA, mediating interactions through cytokines and other inflammatory signals. Cytokines released by gut immune cells can cross the BBB and influence brain activity, often contributing to neuroinflammation [45]. Conversely, stress and CNS disorders can impact gut immunity, affecting intestinal permeability and microbiota balance. This bidirectional influence underscores the immune system’s role in diseases such as IBD and depression [46].

4.
**Microbial pathways**


The gut microbiota produces a variety of metabolites that interact with the brain, influencing neurological functions, gastrointestinal motility, and overall health. Some key metabolites include the following:Short-chain fatty acids (SCFAs): produced from dietary fiber fermentation, SCFAs like butyrate, acetate, and propionate serve as energy sources for colonocytes and modulate BBB integrity and regulate gut motility and immune responses [17,47];Serotonin precursors: gut microbes influence the metabolism of tryptophan, an essential amino acid obtained from dietary sources, which is crucial for serotonin synthesis and impacts mood regulation and intestinal peristalsis [17,48];GABA, other neurotransmitters, and neurotransmitter derivatives: certain bacteria can produce or stimulate the production of neurotransmitters. For instance, *Lactobacillus* produces acetylcholine and γ-GABA; *Bifidobacterium* produces γ-GABA; *Bacillus* and *Escherichia coli* produce norepinephrine, serotonin, and dopamine; and *Streptococcus*, *Enterococcus*, and *Candida* produce serotonin, which directly affects brain function [17,48];Trimethylamines and amino acid derivatives: these metabolites have systemic effects, influencing host metabolism and potentially impacting brain health [47].
5.**Integrated System**

The GBA is an integrated system with feedback loops between its various components. Disruptions in any part of the axis, whether microbial, neural, or hormonal, can propagate through the system, leading to physiological or psychological dysfunctions. For instance, chronic stress can impair vagus nerve function, alter microbiota composition, and exacerbate intestinal inflammation, further reinforcing the interconnectedness of the GBA [31].

### 3.3. Essential Oils and the GBA

EOs are complex mixtures of volatile, aromatic, and hydrophobic liquids characterized by their oily texture and distinctive odor. They are secondary metabolites produced by plants and stored primarily in specialized structures such as secretory cells, epidermal cells, cavities, canals, or glandular trichomes [49]. Derived from various plant tissues—such as flowers, buds, stems, leaves, roots, fruits, bark, resins, or wood—EOs consist mainly of monoterpenes, sesquiterpenes, diterpenes, and their oxygenated derivatives. Additionally, they contain phenols, aldehydes, ketones, esters, alcohols, and hydrocarbons, which contribute to their broad biological activities, including antioxidant, antimicrobial, insecticidal, and other therapeutic properties [50].

The prominent diffusion of advanced techniques for administration and maximized absorption in humans and animals demonstrates the growing attention toward EOs [51]. Nowadays, more and more products on the market containing EOs are conveyed through microencapsulation technology. Microencapsulated EO technology represents a pivotal advancement in encapsulating and delivering bioactive compounds. This technique ensures the preservation and controlled release of essential oils, enhancing their stability and extending their functionality in various applications such as food preservation, health supplements, and pest control. A study [52] demonstrated complex coacervation for encapsulating lavender oil, achieving improved protection and extended application potential. Similarly, Chen et al. [53] showed significant benefits in growth and intestinal health in weaned piglets with microencapsulated oils, highlighting its application in animal nutrition. Moreover, the dual crosslinked microcapsules enhance antioxidant capabilities, improving long-term efficiency and extending food shelf life. Additionally, microencapsulation was shown to increase the antibacterial stability of essential oil by addressing its volatility, ensuring sustained antibacterial action, and slowing food spoilage [54].

#### 3.3.1. Cannabis EOs on GBA

In the context of the GBA, certain EOs, notably those derived from cannabis (e.g., containing cannabidiol (CBD) and tetrahydrocannabinol (THC) (Figure 1), have attracted significant attention for their potential neuroprotective and anti-inflammatory effects.

These oils, containing a large pool of molecules, are believed to interact with the endocannabinoid system (ECS), a critical regulatory network that influences gut and brain functions. The ECS comprises receptors, such as CBR1 and CBR2, that modulate various physiological processes, including immune responses, gut motility, and neurotransmission [55]. These receptors are extensively distributed all over the body. Specifically, CBR1 is predominantly found in the CNS, particularly in regions such as the cortex, basal ganglia, hippocampus, and cerebellum. At the same time, CBR2 is less prevalent in the CNS but primarily located in microglial and vascular elements and can be expressed by neurons under certain pathological conditions [56].

Additionally, CBR1 is notably present in the ENS [57], liver, and vascular, sympathetic, and sensory nerve terminals [58]. These receptors exist because our body produces many lipid-based small molecules called endocannabinoids (ECBs), which are fundamental for cellular homeostasis [59]. Anandamide (AEA) and 2-arachidonoylglycerol (2-AG) are the most well-characterized endocannabinoids, both interacting with CBR1 and CBR2 receptors, albeit with different affinities and physiological roles [60]. AEA and 2-AG are an amide and an ester derived from ω-6 arachidonic acid [59] (Figure 2).

ECBs have been implicated in the pathophysiology of several neuropsychiatric disorders, including schizophrenia [61], major depressive disorder [62], and post-traumatic stress disorder [63]. At the same time, they appear to exert protective effects in other conditions, such as eating disorders, IBS, and age-related neurodegenerative diseases [64].

CBR1 and CBR2 agonists contribute to colitis protection by facilitating intestinal barrier healing. THC is the primary psychoactive compound having a similar affinity towards CBR1 and CBR2 [65]. However, the enteric pathway distribution of CBR1 and CBR2 differs across neuronal subtypes. While both receptor mRNAs are expressed in the whole gut of guinea pigs, only CBR1 mRNA is detected in the myenteric plexus. Additionally, CBR1 plays a role in reducing gastrointestinal motility by relaxing neurogenic circular muscles through cholinergic neurotransmission [66]. Several in vivo studies have shown that cannabinoid receptor agonists, both natural (e.g., THC) and synthetic (e.g., WIN 55,212-2, AM841), decrease gastrointestinal motility by inhibiting smooth muscle contraction and neurotransmitter release [67,68,69]. Furthermore, activation of cannabinoid receptors has been associated with reduced visceral hypersensitivity in models of inflammatory and functional bowel disorders [70]. 

In contrast, CBR2 exhibits immunosuppressive properties relevant to Crohn’s disease and inflammatory bowel syndrome [71]. These findings suggest that CBRs play distinct roles in regulating peristalsis, gastric acid secretion, food intake, and mucosal healing, potentially influencing the pathophysiology of various intestinal diseases, including metabolic disorders, obesity, IBS, IBD, UC, CD, and even gastrointestinal cancer [72]. However, some findings have indicated a simultaneous effect of marijuana consumption on gut bacteria. THC is a key neuromodulatory and psychoactive phytochemical that affects CBRs, influencing gut bacteria and their metabolites [73]. Additionally, patients who frequently smoke marijuana also exhibit oral dysbiosis, with the enriched bacteria in this condition being a major contributor to CNS abnormalities [74].

Nevertheless, a study highlights how an increase in ECB signaling throughout the GBA can reduce gut–barrier permeability with all its consequent positive effects [75]. On the other hand, chronic cannabis consumption has been associated with a decrease in *Prevotella* and an increase in *Bacteroides* within the gut microbiome. A pilot study shows that the reduction in *Prevotella* levels may contribute to systemic mitochondrial dysfunction and a decline in gut SCFAs production, which in turn could be linked to cognitive impairments in cannabis users [76]. Furthermore, a systematic review published in 2023 demonstrated a significant reduction in the abundance of *Prevotella* in patients with Parkinson’s disease [77]. In addition, a separate study using a mouse model revealed that *Prevotella* transplantation promoted neurorehabilitation following traumatic brain injury [78]. Collectively, these findings suggest a beneficial role of *Prevotella* in brain health, although the underlying mechanisms of action remain incompletely understood.

The synergistic interplay between cannabinoids, terpenes (e.g., myrcene for relaxation), and other bioactive compounds in cannabis that enhance its therapeutic efficacy beyond the effect of isolated components is known as the “entourage effect” [79]. Furthermore, some authors demonstrate that whole cannabis extracts exhibit greater anti-inflammatory and antidepressant-like effects in vitro and in vivo, supporting the concept of synergy among phytochemicals [80]. However, the precise mechanisms underlying the effects of cannabis essential oils on the GBA remain to be fully elucidated.

#### 3.3.2. Other EOs Modulating the GBA

Recently, a study clarified how rosemary (*Rosmarinus officinalis* L.) EO, containing more than 16 identified compounds, with major compounds being camphor, cineole, α-pinene, camphene, and α-terpineol [81] (Figure 3), can act as an anxiolytic agent and consequently improve the cognitive function by enhancing the activation and secretion of dopamine [82]. Furthermore, all the molecules within rosemary EO show a wide range of pharmacological effects, including antioxidant, metal-chelating, and anti-inflammatory properties. These same mechanisms also seem to play a role in the potential therapeutic effects of the compounds for Alzheimer’s disease [83].

In Eucalyptus oil, the primary constituents in the leaves responsible for its CNS effects are 1,8-cineole (Eucalyptol) and α-pinene [84]. In addition to its anti-inflammatory and antimicrobial properties, α-pinene inhibits acetylcholinesterase, the enzyme responsible for breaking down the neurotransmitter acetylcholine into choline and acetate. This inhibition increases acetylcholine levels and prolongs its activity in the CNS, thereby supporting memory function [85].

EOs can exert prebiotic-like effects by selectively promoting beneficial bacterial taxa while suppressing pathogenic species. In humanized mice harboring gut microbiota from patients with ischemic heart disease and type-2 diabetes, savory (*Satureja hortensis*), parsley (*Petroselinum crispum*), and rosemary (*Rosmarinus officinalis*) EOs increased *Lactobacillus* populations by 40–60%. This microbial shift correlated with elevated fecal SCFAs—key metabolites supporting colonic health [86].

The antimicrobial specificity of EOs is exemplified by oregano oil (OEO), which reduced *Escherichia coli* colonization in pig jejunum; in the same study, OEO was demonstrated to increase the expression of tight junctions (which improves the intestinal barrier integrity) as well as a reduction in the proinflammatory cytokines [87].

A new *Thymus vulgaris* L. solid essential oil (SEO) formulation exerted multiple beneficial effects, like a relaxant activity on the K^+^-depolarized intestinal smooth muscle of ex vivo ileum and colon of a guinea pig and antimicrobial activity against different strains (*Staphylococcus aureus*, *Streptococcus pyogenes*, *Pseudomonas aeruginosa*, *Escherichia coli*, *Salmonella Typhimurium*, and *Candida albicans*), similarly to liquid oil, with activity against pathogens but not commensal strains (*Bifidobacterium breve* and *Lactobacillus fermentum*) [88]. Thymol, a significant component of thyme essential oil, may provide neuroprotective benefits while alleviating cognitive deficits, depressive-like behaviors, and impairments in learning and memory in rodents [89].

Silexan®, a proprietary EO obtained through steam distillation of *Lavandula angustifolia* Miller flowers, has shown significant anxiolytic effects, which seem to be mediated by a voltage-dependent calcium channel mechanism. As a potential mechanism underlying its effects on neuroplasticity, Silexan® induces notable phosphorylation of protein kinase A, thereby initiating the phosphorylation of cAMP response element-binding protein, a key downstream target common to all antidepressant drugs [90].

These findings illustrate how EOs act through diverse and sometimes complementary pathways, suggesting potential tailored interventions depending on the targeted outcome, whether cognitive enhancement, stress reduction, or gut barrier reinforcement. Notably, some EOs, such as rosemary and thyme, exert multifaceted actions by modulating microbial and neurochemical processes, while others, like lavender or eucalyptus, may primarily influence central neurotransmission. This functional differentiation opens the door to formulation strategies that combine specific EOs to maximize synergistic effects on the GBA. Additionally, the feasibility of incorporating these compounds into nutraceuticals, functional foods, or microencapsulated systems enhances their translational potential. Although the preclinical data are promising, further work is needed to define optimal dosages, administration routes, and long-term safety profiles to support their integration into preventive or therapeutic protocols.

## 4. Discussion

The literature reviewed in this manuscript confirms the growing scientific interest in essential oils as modulators of the GBA, particularly through their impact on microbiota composition, neurotransmitter activity, and neuroinflammatory pathways. The various EOs examined, including those derived from *Cannabis sativa*, *Rosmarinus officinalis*, *Lavandula angustifolia*, *Eucalyptus globulus*, *Thymus vulgaris*, and *Origanum vulgare*, each exhibit unique profiles of action. This heterogeneity suggests that specific EO combinations or targeted formulations could be strategically employed to influence different components of the GBA. Furthermore, their ability to modulate neuroactive metabolite production and immune signaling supports their potential as complementary therapeutic tools in neurological, psychiatric, and gastrointestinal disorders. These findings provide a solid foundation for discussing both the current limitations and the translational potential of EO-based interventions.

It has been observed that various substances can impact the gut–brain axis. However, in light of the most recent evidence, it is still impossible to determine whether there is a predominant flow of information from the brain to the gut or *vice versa*.

Another critical question is whether the changes in microbiota composition observed in different neurological patients are an effect of the disease or if they are the cause of the pathology in the brain. Other hypotheses suggest a direct involvement of altered microbiota in the development of the disease, which, in turn, could further modulate the microbiota during its progression.

It can be stated with certainty that there is a correlation between gut alterations and certain neurological, psychological, or neurodegenerative disorders [91]. In most cases, the manifestation of the disease is linked to an imbalance in microbiota, which can be effectively addressed with the substances discussed in this review through the different mechanisms of action mentioned earlier.

Although studies show that the phytocannabinoids present in cannabis EOs have a short-term positive impact on health through various actions on GBA, this evidence appears to reverse with chronic use [92].

Current evidence suggests that certain EOs exert a more pronounced effect on the central nervous system, while others primarily promote beneficial modifications in gut microbiota composition (Table 1). Nonetheless, both mechanisms contribute synergistically to mitigating diseases whose etiopathogenesis is now recognized as associated with GBA dysregulation.

A critical frontier yet to be explored is identifying microbiota alterations (biological imprint) associated with the possible future onset of a specific disease. This would enable early diagnosis, meaning the ability to diagnose a condition that has not yet manifested symptoms solely based on intestinal alterations.

The objective is to identify natural compounds, including plant extracts, bioactive molecules derived from food waste, and EOs, that can synergistically modulate gut microbiota imbalances and possibly alterations in gut functions, while addressing mental health disorders and neurodegenerative diseases. This review highlights the importance of developing EO-based strategies targeting the microbiota and CNS to support integrative therapeutic approaches in GBA-associated disorders. Clarifying EO-based interventions’ mechanistic pathways and clinical efficacy will be pivotal to unlocking their full therapeutic potential in gut–brain-related disorders.

## 5. Conclusions and Future Perspectives

The findings discussed throughout this review offer a promising foundation for future integrative therapeutic applications.

This narrative review highlights the therapeutic potential of EOs in modulating the GBA, emphasizing their dual action on gut microbiota composition and central nervous system function. A significant strength of this study lies in its integrated and multidisciplinary approach, which considers natural compounds as promising multitarget agents for managing gastrointestinal, neurological, and psychiatric disorders. However, several limitations persist, including the scarcity of clinical trials and meta-analyses, the absence of long-term observational studies, the high variability in both the qualitative and quantitative composition of essential oils, and the inability to determine the directionality of gut–brain communication. Differences in the concentrations of active compounds, such as CBD and THC, can significantly influence the observed biological effects, complicating the interpretation and comparison of study outcomes.

Further research is necessary to clarify the mechanistic pathways involved and assess the long-term efficacy of EO-based interventions in clinical settings. Among the EOs discussed, compounds derived from cannabis, rosemary, lavender, and oregano appear particularly promising and deserve focused investigation.

Ultimately, a deeper understanding of EO-mediated modulation of the GBA could lead to novel, natural therapeutic strategies targeting both brain and gut health. In this regard, exploring personalized EO-based interventions adapted to specific microbiota profiles may enhance clinical outcomes. Moreover, integrating essential oils into functional foods or nutraceutical formulations could increase patient compliance and expand their preventive applications. This approach also contributes to reducing healthcare costs and improving patient quality of life by promoting sustainable, natural, and individualized prevention strategies. Early identification of gut dysbiosis patterns associated with neuropsychiatric or neurodegenerative conditions might also enable targeted EO-based therapies before symptom onset, representing a crucial step toward predictive and precision medicine.

## Figures and Tables

**Figure 1 nutrients-17-01578-f001:**
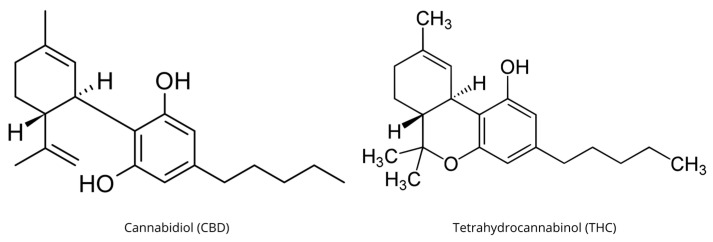
Chemical structure of the two principal phytocannabinoids: cannabidiol (CBD) and tetrahydrocannabinol (THC).

**Figure 2 nutrients-17-01578-f002:**
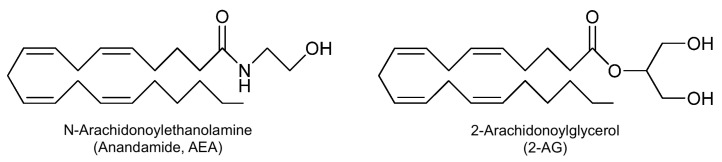
Chemical structures of AEA and 2-AG.

**Figure 3 nutrients-17-01578-f003:**
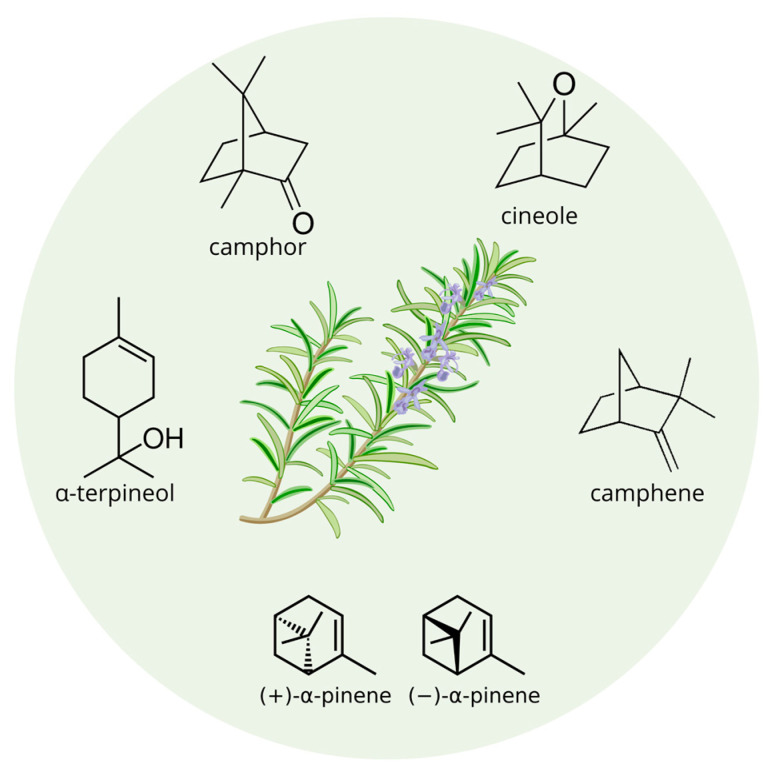
Main components of rosemary EOs.

**Table 1 nutrients-17-01578-t001:** EOs’ main effects summary.

EOs	Main Effects	References
Rosemary	Anxiolytic, ↑ cognitive function, antioxidant, metal-chelating, and anti-inflammatory.	[82]
Eucalyptus	↑ Memory function, anti-inflammatory, and antimicrobial.	[84,85]
Oregano	↑ Intestinal barrier integrity, and ↓ proinflammatory cytokines.	[87]
Thymus	Intestinal smooth muscle relaxation, antimicrobial, and neuroprotective.	[88,89]
Silexan®	Anxiolytic effects.	[90]

Up arrow: increase, down arrow: reduction.

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
