# Peer review of "Targeting the Gut–Brain Axis with Plant-Derived Essential Oils: Phytocannabinoids and Beyond"

_nutrients, 2025, doi:10.3390/nu17091578_

Round 1

Reviewer 1 Report

Comments and Suggestions for Authors

The article is a comprehensive review of the literature on the effects of essential oils (EOs), including phytocannabinoids, on the gut-brain axis (GBA). This is a current and interdisciplinary topic, combining gut microbiology, neurobiology, phytochemistry, and pharmacology. The work touches on one of the key areas of research — the relationship between gut microbiota and the nervous system and the impact of natural substances on mental and intestinal health.

The introduction presents a well-founded motivation for the study and outlines the clinical and biological context. The review sections are logically organized: from the anatomy of GBA, through the mechanisms of communication, to a detailed discussion of individual oils. Manuscript contains descriptions of the effects of both cannabinoids (CBD, THC) and other plant oils (lavender, eucalyptus, oregano, rosemary), which gives the work a wide application dimension. Manuscpript focuses on the role of CBR1/CBR2 receptors, neurotransmitters, HPA, blood-brain barrier, microbiota and immune system in the mechanisms of action. Also, the authors rightly point out the negative effects of long-term use of hemp oil, e.g. microbiota and cognitive dysfunctions.

Major remarks:

A few places contain placeholders of the type “Error! Reference source not found.” (e.g. lines 205, 226, 269), which indicates an unfinished editing stage and may disorganize the reception of the content.

Figures (e.g. CBD, THC, AEA, 2-AG structures on pp. 5 and 6 and rosemary oil diagram on p. 7) are correct, but they lack page numbers and references in the main text.

There is no summary table comparing the properties of different EOs in the context of GBA.

Some statements are not clearly supported by clinical studies or meta-analyses, e.g. the effect of Prevotella reduction on mitochondrial and cognitive functions (line 255–257).

The discussion (p. 8) could include a more critical assessment of the current limitations of knowledge and more specific suggestions for directions for future research.

Author Response

We thank Reviewer 1 for our manuscript's positive assessment and constructive suggestions. Please find below our responses to each of the comments.

Comments 1: A few places contain placeholders of the type “Error! Reference source not found.” (e.g. lines 205, 226, 269), which indicates an unfinished editing stage and may disorganize the reception of the content.

Response 1: Thank you for your observation. All the errors of the type “Error! Reference source not found.” have been corrected in the indicated sections of the text.

Comments 2: Figures (e.g. CBD, THC, AEA, 2-AG structures on pp. 5 and 6 and rosemary oil diagram on p. 7) are correct, but they lack page numbers and references in the main text.

Response 2: Thank you for the comment. All figures (e.g., structures of CBD, THC, AEA, 2-AG, and rosemary EO components) are now explicitly referenced in the main text (e.g., “as shown in Figure 1”) to improve clarity [lines 257, 277, 324]. Regarding page numbering, we understand this is a formatting step handled during the journal production process.

Comments 3: There is no summary table comparing the properties of different EOs in the context of GBA.

Response 3: Thank you for your valuable comment. A summary table comparing the properties of different essential oils in the context of the gut-brain axis has now been introduced in the discussion section of the manuscript [line 410].

Comments 4: Some statements are not clearly supported by clinical studies or meta-analyses, e.g. the effect of Prevotella reduction on mitochondrial and cognitive functions (line 255–257).

Response 4: Thank you for your insightful comment. The text has been revised accordingly, and two additional studies have been introduced to support the previously stated claims regarding the effects of Prevotella. [lines 305-313].

Comments 5: The discussion (p. 8) could include a more critical assessment of the current limitations of knowledge and more specific suggestions for directions for future research.

Response 5: A Conclusions and Future Perspectives section has been added, highlighting the current limitations in knowledge and providing more specific suggestions for future research directions. [lines 424-449].

Reviewer 2 Report

Comments and Suggestions for Authors

Firstly, the authors will need to work to decrease the similarities of their manuscript with other published investigations. The current similarity index is too high.

The type of review should be expressed in the title, in the abstract, and in the whole manuscript.

The abstract should be structured and revised. It is too long (the word limit is 250). The methods are not mentioned (searched databases, search strategy, inclusion/exclusion criteria, searched keywords, etc.…). Some results should be indicated quantitatively.

The Introduction can be improved, and the study’s novelty should be better justified.

A Methods section is missing. See the example of section 2 of this paper: https://www.mdpi.com/2304-8158/10/6/1175

The manuscript is well written, and relevant data are provided. However, the Discussion section can be improved, and some tables should be included to describe better/summarize the relevant studies included in this review.

The authors should also discuss the study’s main limitations and strengths.

A Conclusion section and Future Perspectives are missing.

Author Response

We thank Reviewer 2 for the detailed and helpful comments that allowed us to improve the clarity and completeness of our manuscript. Below are our responses.

Comments 1: Firstly, the authors will need to work to decrease the similarities of their manuscript with other published investigations. The current similarity index is too high.

Response 1: We acknowledge the reviewer’s concern regarding textual similarity. We carefully reviewed the manuscript and have rewritten several sections to reduce similarity with previously published works, especially in the introduction and background paragraphs. We also ensured all references were correctly cited and the wording reflected the original synthesis. We are confident that the revised version significantly reduces the similarity index, but we remain open to further suggestions if needed.

Comments 2: The type of review should be expressed in the title, in the abstract, and in the whole manuscript.

Response 2: Thank you for the suggestion. We have clarified the type of article by explicitly stating that it is a narrative review above the title, in the Methods section, and in the Conclusions and Future Perspectives section of the manuscript. [lines 125, 137, 427].

Comments 3: The abstract should be structured and revised. It is too long (the word limit is 250). The methods are not mentioned (searched databases, search strategy, inclusion/exclusion criteria, searched keywords, etc.…). Some results should be indicated quantitatively.

Response 3: Thank you for your helpful comment. The abstract has been revised to comply with the journal’s word limit (≤250 words) and is now structured into the standard format (Background, Methods, Results, Conclusions). We have included a concise description of the literature search strategy, specifying the databases consulted, keywords used, and general inclusion/exclusion criteria. Additionally, quantitative data from the screening process (number of articles initially identified and selected) have been integrated into the Methods section of the manuscript. The revised abstract reflects these updates accordingly. [lines 15-35].

Comments 4: The Introduction can be improved, and the study’s novelty should be better justified.

Response 4: We agree, the introduction has been improved according to clarify also the study’s novelty as it was suggested. [85-119]

Comments 5: A Methods section is missing. See the example of section 2 of this paper: https://www.mdpi.com/2304-8158/10/6/1175.

Response 5: We thank the referee for the valuable example provided. In response to the suggestion, a Methods section has been added to the manuscript, including quantitative results from the literature screening process. [lines 124-139].

Comments 6: The manuscript is well written, and relevant data are provided. However, the Discussion section can be improved, and some tables should be included to describe better/summarize the relevant studies included in this review.

Response 6: Thank you for your valuable comment. A summary table comparing the properties of different essential oils in the context of the gut-brain axis has now been introduced in the discussion section of the manuscript. [line 410].

Comments 7: The authors should also discuss the study’s main limitations and strengths.

Response 7: The main strengths and limitations of the study have been addressed and are now integrated into the newly added Conclusions and Future Perspectives section, which was previously missing from the manuscript. [lines 424-449].

Comments 8: A Conclusion section and Future Perspectives are missing.

Response 8: Thank you for your valuable suggestion. As recommended, the manuscript has now included a Conclusion and Future Perspectives section. [lines 424-449].

Round 2

Reviewer 1 Report

Comments and Suggestions for Authors

The manuscript has been modified accordingly to suggestions. 

Author Response

                                                                                                                    Bologna, 28 April 2025

Revision of Manuscript ID: nutrients-3593907

“Targeting the Gut-Brain Axis with plant-derived Essential Oils: 2 phytocannabinoids and beyond”

by: Luca Camarda, Laura Beatrice Mattioli, Ivan Corazza, Carla Marzetti and Roberta Budriesi

Dear Editor,

We have carefully considered the comments and are sincerely grateful for their valuable feedback, which has helped us further improve our manuscript's quality. The paper has been extensively revised to address all the points raised. Below, we provide a detailed, point-by-point response to each comment.

We hope that the revised version will be suitable for publication.

Thank you once again for your time and consideration. We look forward to your feedback.

                                                                                                         Sincerely

                                                                                               Luca Camarda Dr.

1) I believe in this section it is necessary to mention one of the most frequent symptoms encountered in patients suffering from disorders of the gut-brain interaction, which is visceral pain; it seems relevant to include a brief paragraph on this because essential oils also exert beneficial effects on this problem (i.e., mint oil is used specifically to address this); please consider including also a mention to these effects when describing the different essential oils you have included in your review.

We thank the reviewer for this valuable and insightful suggestion. Following your recommendation, we have now included a specific paragraph highlighting that one of the most frequent symptoms encountered in patients suffering from disorders of the gut-brain interaction is visceral pain. We have also discussed how essential oils, particularly peppermint oil, can exert beneficial effects on this symptom by reversing visceral hypersensitivity, representing a safe and effective therapeutic approach. This information has been incorporated into the manuscript, and we have cited the following reference to support it: Botschuijver S. et al., Neurogastroenterology and Motility, June 2018. [Botschuijver S, Welting O, Levin E, Maria-Ferreira D, Koch E, Montijn RC, Seppen J, Hakvoort TBM, Schuren FHJ, de Jonge WJ, van den Wijngaard RM. Reversal of visceral hypersensitivity in rat by Menthacarin®, a proprietary combination of essential oils from peppermint and caraway, coincides with mycobiome modulation. Neurogastroenterol Motil. 2018 Jun;30(6):e13299. doi: 10.1111/nmo.13299. Epub 2018 Jan 31. PMID: 29383802.].

2) Please check if this reference is correct here because it seems to be focused on Alzheimer's disease and not on inflammatory bowel disease and metabolic disorders.

We thank the reviewer for the insightful comment. Upon careful review, we agree that the previous reference [38] was not appropriate for supporting the statement regarding IBD and metabolic disorders. We have now replaced it with a more suitable reference: [Vindigni SM, Zisman TL, Suskind DL, Damman CJ. "The intestinal microbiome, barrier function, and immune system in inflammatory bowel disease: a tripartite pathophysiological circuit with implications for new therapeutic directions." Therapeutic Advances in Gastroenterology, 2016]. This reference discusses in detail the disruption of gut barrier function, dysbiosis, and their roles in the pathogenesis of inflammatory bowel disease and related metabolic conditions.

3) The vagus nerve has also a key influence on inflammation, please consider adding a mention here to this effect.

However, the vagus nerve is not the only extrinsic nerve pathway that connects to gastrointestinal tract with the central nervous system, there are other extrinsic nerve pathways that are involved, particularly in visceral pain, such as the pelvic nerves and the splachnic nerves. These nerve pathways should also be mentioned, these are essential to explain many disorders of the gut-brain interaction...

We thank the reviewer for these very helpful observations. We have now modified the section to better highlight the role of the vagus nerve not only in gut motility and secretion but also in modulating systemic inflammation. Moreover, following the suggestion, we have added a mention of other extrinsic nerve pathways, including the pelvic and splanchnic nerves, which are particularly involved in the transmission of visceral pain and are essential for understanding disorders of the gut-brain interaction.

4)You should mention that they also modulate gastrointestinal motility, with relevant references.

We thank the reviewer for the valuable suggestion. We have now amended the manuscript to explicitly mention that microbial metabolites, including SCFAs and serotonin precursors, also regulate gastrointestinal motility in addition to their effects on neurological functions and immune responses. Relevant references [17,46,47] have been maintained.

5) I think this should be changed to GABA and other neurotransmitters and neurotransmitter derivatives

We appreciate the reviewer's suggestion. We have revised the heading to "GABA, other neurotransmitters, and neurotransmitter derivatives" to better reflect the diversity of microbial-derived neuroactive compounds described in the manuscript.

6) Please change to “and other therapuetic properties”

Done

7) Different concentrations of??

I guess there are many variations of essential oils and not only the qualitative but also the quantitative composition of these liquids would influence the observed effects. Indeed, I think this should be mention as a general limitation of these studies (and possibly target for future research), in the discussion and possibly in the conclusion section.

We thank the reviewer for the insightful comment. As suggested, we have added a statement in the Discussion section (line 451) addressing the variability in both the qualitative and quantitative composition of essential oils. We specifically highlighted how differences in the concentrations of active compounds, such as cannabidiol (CBD) and tetrahydrocannabinol (THC), may significantly influence the observed biological effects. This limitation has been acknowledged as a critical factor complicating the interpretation of study outcomes and has been indicated as an important target for future research.

8) it might be interesting for the reader to add some references regarding the in vivo effects of cannabinoid agonists on motility (selective and non-selective, natural and synthetic).

Additionally, their effects on visceral pain should also be briefly described.

We thank the reviewer for the helpful suggestion. In response, we have expanded the paragraph by adding a brief description of the in vivo effects of cannabinoid receptor agonists, including natural compounds like THC and synthetic compounds such as WIN 55,212-2 and AM841, on gastrointestinal motility and visceral pain modulation. In addition, we included clinical evidence from human studies demonstrating the impact of cannabinoid mechanisms on gut motor functions. The following references have been added: Izzo and Sharkey (2010), Guindon and Hohmann (2008), Abalo et al. (2015), and Camilleri (2018).

9) please check the use of abbreviations throughout the manuscript: once defined at first use, please be consistent

We thank the reviewer for the suggestion. We carefully checked the use of abbreviations throughout the manuscript to ensure consistency. In particular, we have now defined and standardized the abbreviations for IBS (Irritable Bowel Syndrome), IBD (Inflammatory Bowel Disease), UC (Ulcerative Colitis), and CD (Crohn’s Disease) at their first occurrence in the text. We have added as suggested “and even gastrointestinal cancer”.

10) Please change “Additional” to “Other”

Done

11) in this table, it is not clear what effects are due to oregano and thymus, because of the layout of the table... would it be possible to include capital letter only on the first word of the "main effects" exerted by the particular essential oil? Maybe this would help differentiate among them

We thank the reviewer for the useful observation. We have revised the layout of the table by using capital letters only for the first word of each "Main effects" description, in order to improve clarity and to better associate each set of effects with the corresponding essential oil. Additionally, we have added horizontal lines to the table to further enhance its readability.
